# Determining the Spectral Requirements for Cyanobacteria Detection for the CyanoSat Hyperspectral Imager with Machine Learning

**DOI:** 10.3390/s23187800

**Published:** 2023-09-11

**Authors:** Mark W. Matthews, Jeremy Kravitz, Joshua Pease, Stephen Gensemer

**Affiliations:** 1CyanoLakes (Pty) Ltd., Cherrybrook, NSW 2126, Australia; 2NASA Postdoctoral Program, Oak Ridge Associated Universities, NASA Ames Research Center, Moffett Field, CA 94035, USA; jeremy.kravitz@nasa.gov; 3Bay Area Environmental Research Institute, Moffett Field, CA 94035, USA; 4NASA Ames Research Center, Moffett Field, CA 94035, USA; 5CSIRO Manufacturing, Urrbrae, SA 5064, Australia; joshua.pease@csiro.au (J.P.); stephen.gensemer@csiro.au (S.G.)

**Keywords:** chlorophyll-a, cyanobacterial blooms, CyanoSat, hyperspectral, linear variable filter, machine learning, phycocyanin, spectral requirements

## Abstract

This study determines an optimal spectral configuration for the CyanoSat imager for the discrimination and retrieval of cyanobacterial pigments using a simulated dataset with machine learning (ML). A minimum viable spectral configuration with as few as three spectral bands enabled the determination of cyanobacterial pigments phycocyanin (PC) and chlorophyll-a (Chl-a) but may not be suitable for determining cyanobacteria composition. A spectral configuration with about nine ideally positioned spectral bands enabled estimation of the cyanobacteria-to-algae ratio (CAR) and pigment concentrations with almost the same accuracy as using all 300 spectral channels. A narrower spectral band full-width half-maximum (FWHM) did not provide improved performance compared to the nominal 12 nm configuration. In conclusion, continuous sampling of the visible spectrum is not a requirement for cyanobacterial detection, provided that a multi-spectral configuration with ideally positioned, narrow bands is used. The spectral configurations identified here could be used to guide the selection of bands for future ocean and water color radiometry sensors.

## 1. Introduction

Cyanobacteria and harmful algal blooms (HABs) have garnered significant attention due to their potential ecological and health impacts on inland waters [1]. These blooms can introduce toxins that threaten aquatic life, contaminate drinking water supplies, and disrupt recreational activities [2,3,4]. Accurate, timely, and cost-effective monitoring methods are paramount in managing these outbreaks and mitigating their adverse effects. With advancements in satellite sensor technology, remote sensing has emerged as a potent tool for the surveillance and quantification of HABs [5]. Determining the optimal spectral resolution of these sensors is pivotal in maximizing their accuracy and efficiency.

Despite the wealth of studies detecting cyanobacterial presence and pigment concentrations using optical satellite sensors [6,7,8,9,10,11], there are currently no studies that directly determine the minimum spectral requirements for cyanobacteria discrimination. The impact of the spectral resolution of existing and planned satellite instruments on PC estimation has already been examined in [12,13], and the characteristic spectral features of cyanobacterial blooms have been investigated in [14]. This study contributes towards a fresh understanding of the spectral requirements for cyanobacteria detection by examining various spectral configurations for a novel optical imager, CyanoSat, using an ML approach. CyanoSat is a custom hyperspectral imager under design by the Commonwealth Scientific and Industrial Research Organization (CSIRO, Australia’s national science agency) to monitor and detect cyanobacterial blooms in fresh and near-coastal waters on a microsatellite payload. CyanoSat is designed to capture a heavily oversampled spectrum through overlapping spectral channels, which can be arbitrarily averaged into discrete spectral bins to improve the signal-to-noise ratio (SNR). The goal of this study is to assess various spectral configurations for CyanoSat with the aim of optimally configuring the instrument for cyanobacteria discrimination, detection, and quantification. The study objectives are:Determine the relative importance of each of the instrument’s spectral channels for quantitative cyanobacteria detection with a view to prioritizing bands for download or grouping.Assess the performance of the CyanoSat and various minimal spectral configurations for determining the concentration of pigments Chl-a, PC, and cyanobacteria composition.Assess parameter retrieval performance with top-of-atmosphere (TOA) data types that avoid the need for error-prone atmospheric correction of CyanoSat data.Assess whether there is any advantage to a narrower FWHM of 8 nm compared to the nominal 12 nm FWHM CyanoSat configuration.

We use an as realistic as possible, simulated (synthetic) dataset that uniquely handles variability in the phytoplankton community composition (variable cyanobacteria:algae ratios), encompasses the full range of likely cyanobacterial biomass (Chl-*a* up to 1000 s µg/L), models Chl-*a* fluorescence for both eukaryotic algae and cyanobacteria separately, and takes into consideration the cyanobacterial phycobiliproteins (phycocyanin, phycoerythrin) and algal pigments [12]. This makes it an ideal candidate for assessing requirements for spectral configurations for cyanobacteria detection using CyanoSat. The synthetic dataset also includes a diverse range of water types and atmospheric optics including effects from stray light from vegetation (the adjacency effect) which is known to significantly affect the remotely sensed signal over small inland water bodies (lakes and rivers) [15]. The analysis is performed using water-leaving reflectance (*R_rs_*), TOA reflectance (TOAR), and partially corrected bottom-of-Rayleigh reflectance (BRR) data types. This provides an initial assessment of the feasibility of deriving cyanobacteria detection products from CyanoSat data with or without atmospheric correction, from an ML perspective.

## 2. Materials and Methods

### 2.1. Description of the CyanoSat Imager

The CyanoSat imager is a visible-to-near-infrared imaging spectrometer that utilizes a custom-designed F/4.6 three-mirror anastigmat telescope and a compact linear variable filter (Figure 1). To ensure compatibility with cube-sat class satellite buses the payload has a volume envelope of 100 mm × 100 mm × 300 mm and a weight of 2.7 kg. The instrument is designed to have a nominal spatial resolution of 50 m in low-Earth orbit, corresponding to an across-track swath width of 24 km. The spectral range of the instrument is 500 to 780 nm, with a native spectral sampling interval of approximately 1 nm. Given the configuration of the imager, the nominal 12 nm FWHM spectral bands overlap considerably with one another, leading to a spectral smoothing effect (Figure 2). A coaxial monochromatic sensor with a center wavelength of 850 nm provides ancillary spectral information for sun-glint correction. The instrument’s on-board processing capabilities enable data size reduction through spectral binning, image registration, and lossless compression.

### 2.2. Synthetic Dataset

A full description of the synthetic dataset is found in [12]. In-water radiative transfer simulations were performed using EcoLight^®^ (Numerical Optics) and a custom additive bio-optical model with phytoplankton and dissolved and mineral components, with distributions constrained to known optical water types [16]. Roughly 70,000 *R_rs_* spectra were modeled with paired in-water inherent optical properties and biogeophysical data. A functional data analysis [17] approach was taken to identify distinct optical clusters with respect to spectral information. The reflectance spectra were deconvolved into β-spline representations using 26 cubic basis functions and clustering was performed using the basis coefficients. This resulted in 13 distinct clusters similar to those described for natural waters in [16]. Full descriptions of each optical cluster can be found in [12], but briefly, the 13 clusters were condensed into seven manually defined optical water types which are more ecologically tangible. These water types range from oligotrophic waters (“Case 1”) to mildly blooming mixed eutrophic waters, highly scattering waters where inorganic sediment dominates the optical signal, highly absorbing waters where organic color dominates the optical signal, and both eukaryotic and cyanobacteria-dominated hypereutrophic waters including floating cyanobacteria scum conditions. 

The cyanobacteria:algae ratio (CAR) was used to scale the phytoplankton Chl-*a* specific inherent optical properties (SIOPs) to simulate mixed cyanobacterial–algal populations. Cyanobacterial SIOPs were generated using a two-layered coated sphere model [18] and included data from *Microcystis aeruginosa, Anabaena, Aphanizomenon,* and *Nodularia* species [6]. The algal component was modeled from carotenoid-containing algae (*Dinoflagellate* or *Diatom* sp.) dominated by Chl-*a* and the carotenoid pigments, fucoxanthin and peridinin. The cyanobacterial PC concentration for each simulated spectrum was estimated from the total phytoplankton absorption coefficient at 620 nm after removal of the effects for chlorophylls *a*, *b*, and *c* absorption [19]. A variable PC-specific absorption coefficient at 620 nm parameterized using the CAR and the PC:Chl-*a* ratio was used to calculate the final PC concentration. Features related to phytoplankton Chl-*a* fluorescence are diagnostic for cyanobacterial blooms [7,8] and require careful modeling. The simulations improve existing models of phytoplankton fluorescence in the following ways: accounting for the non-uniform distribution of Chl-*a* between photosystems I and II in cyanobacteria; including the often-neglected Chl-*a* fluorescence emission peak near 730 nm; accounting for mixed cyanobacterial–algal populations; and varying the fluorescence quantum yield. The result is more realistic representations of diagnostic fluorescence-related reflectance features for cyanobacterial blooms.

Modtran^®^ (Spectral Sciences Inc., Burlington, MA, USA) radiative transfer code was used to propagate *R_rs_* to TOAR. Realistic atmospheric column properties were randomly varied and derived from the NASA AERONET database [20], with two atmospheres being modeled for each spectrum, including randomly varied stray light from adjacent “green grass”. The synthetic dataset represents a more accurate representation of TOAR by accounting for stray light effects, which are significant in the near-infrared (NIR). From TOAR, the Rayleigh-corrected reflectance (or BRR) was computed. The partial atmospheric correction removes the effects of gaseous absorption while neglecting the effects of aerosols (e.g., smoke and dust) that are highly variable and unpredictable, even for fixed sites. It is relevant to assess spectral configurations for TOAR and BRR data types because atmospheric correction remains error-prone for cyanobacterial and algal blooms and other complex water types. Thus, it is important to identify band configurations that can be optimally applied with uncorrected or partially corrected data types. The simulated dataset consisted of more than 260,000 TOA spectra.

### 2.3. Parameter Retrieval Using Machine Learning

After [12], we have employed an artificial neural network ML algorithm called the multi-layer perceptron (MLP) algorithm to derive estimates of CAR, Chl-*a*, and PC. For this problem, the MLP was shown previously to give the best performance when compared with alternative ML approaches. A detailed overview of how the algorithm was configured is found in [12] but briefly described here. First, the hyperspectral synthetic data were resolved to the different hypothetical band configurations using the sensor spectral response functions. The MLP model used for this study was composed of an input layer, which accepted the visible and near-infrared channel reflectances of the specific sensor configuration, followed by five hidden layers of 100 neurons each and an output layer that included a singular water parameter of choice. The activation function, which maps the summed weighted inputs to the output of the neuron, was performed using Rectified Linear Units (ReLu). The Adam optimization algorithm was used to iteratively update network weights based on training data and acts as a more functional extension of stochastic gradient descent, where the loss function used was mean absolute error (MAE). Several feature transformations were performed on the data before training. The input channel reflectances were log-transformed, as were output Chl-*a* and PC concentrations. Input reflectances were then standardized by removing the mean and scaling to unit variance. The CAR data were untransformed. The dataset was split into 80% for training, k-fold cross-validation for five folds was used to avoid sampling bias, and 20% of the data were used for the validation and application of statistical performance metrics. The correlation between the values estimated by the MLP algorithm and the actual values from the labeled synthetic dataset was used to assess the performance of the various spectral configurations of the CyanoSat instrument. K-fold cross-validation was used where the synthetic dataset was split into 80% training and 20% testing for five folds to avoid sampling bias. The performance metrics used were the correlation coefficient, R^2^, the Mean Absolute Percentage Error (MAPE), and the Log-Transformed Root Mean Square Error (RMSELE) (see [12] for details). All data manipulation, processing, and training were performed in Python and Tensorflow.

### 2.4. Feature Attributions Scores

To understand how individual channels contributed to the retrieved bio-physical quantities, we used a perturbation-based approach [21]. We perturbed the reflectance of each spectrum at each wavelength in our test set and measured how this perturbs the outputs of our retrieval results. The sensitivity to those perturbations will give us an overall understanding of which parts of the spectrum carry the most information for our retrieval objective. Specifically, we used the feature importance permutation algorithm defined in [22] (Algorithm 1):


**Algorithm 1**



$e_{orig\;}=L(y,\hat{f}\;(X))\;\;$

For each feature j ∈ {1,...,p} do:
-Randomly shuffle column j of dataset X to generate permuted feature matrix $X_{perm}$-Estimate error $e_{perm}=L(Y,\;\hat{f}\;(X_{perm})$) based on the predictions of permuted data-Calculate feature importance attribution as ${FI}_{j}=e_{perm}-e_{orig}$Sort features by descending FI


Where $\hat{f}$ is your trained model, X is your feature matrix, y is your target vector, and $L(y,\hat{f})$ is your error measure. When randomly shuffling column j, you break the relationship between the feature and the true outcome. This measures how much the model’s performance decreases when this feature’s information is disrupted, which is interpreted as a measure of the feature’s importance. R^2^ was used as the error metric for feature permutation. Other error metrics were tested; however, they showed quite similar results, and thus R^2^ was used due to its interpretability in regression problems and sensitivity to model perturbations. Future work could potentially take a band-wise average of multiple error metrics, but this would vastly increase computation time and would not be necessary for a theoretical exploration such as this.

Feature permutations on spectral data come with certain caveats. Permutation importance functions under the assumption that each feature (or band in this case) is independent of one another, whereas with spectral data, adjacent bands might be correlated, and feature importance could be misestimated or misinterpreted. To examine this effect, tests using all channels showed very little perceptible difference between adjacent channels (at an effective 1 nm incremental spacing) in attribution scores, and product retrieval performance using all 300 channels was nearly identical to that using every second or third channel (200 or 100 channels). By selecting every third channel, run times were reduced by two-thirds without any degradation in product retrieval. Thus, the analysis was performed using every third channel providing a nominal spectral resolution of 3 nm with 100 bands. In addition to “Full” spectral resolution, consisting of every third channel or 100 bands, we determined two other spectral configurations to identify optimal and reduced minimum viable band scenarios. The first spectral configuration, called “P1”, consists of nine attribution peaks. This number was chosen after experimenting with different configurations and finding that nine bands is close to the optimal spectral configuration that enables product retrieval with almost the same accuracy as Full resolution but using significantly fewer bands. Very little difference was found in performance for 12, 15, 20, or more bands. Secondly, a “P2” configuration was chosen consisting of the three primary spectral attribution peaks. Three bands were decided on after noticing that three broad regions of importance could be defined from the feature importance plots for each product. P2 represents the minimum viable spectral configuration that enables estimation of the product under investigation, albeit with larger errors than the Full and P1 configurations. Product retrievals were performed using all three spectral configurations.

## 3. Results

### 3.1. Band Configurations Based on Feature Attributions

#### 3.1.1. Band Positions for *R_rs_*

The attribution scores in each of the 100 spectral bands for all three products and data types are shown in Figure 3. The spectral bands for the P1 and P2 configurations identified through peak attribution scores are shown in Table 1. First, we assess band selection for *R_rs_*, neglecting atmospheric effects. It is apparent that there are three primary spectral regions that are important for Chl-*a* estimation. These are in three broad channels from 575 to 625 nm, 660 to 710 nm, and 725 to 775 nm, wherein the peak attribution scores are located. These features relate to, in order, (1) the green peak and the maximum absorption of accessory pigments; (2) the Chl-*a* absorption maximum, fluorescence, and the red edge; and (3) particulate scattering in the NIR from particles including phytoplankton. This can be narrowed down to only three spectral bands for the P2 configuration, which are 590, 674, and 749 nm.

For PC, the three most important spectral regions are from 525 to 575 nm, 610 to 640 nm, and 660 to 690 nm. These features relate to the location of (1) accessory pigment absorption, (2) the location of the PC absorption maximum, and (3) Chl-*a* fluorescence and absorption features. This can be narrowed down to a P2 configuration with three bands at 548, 626, and 680 nm. Clearly, differences in pigmentation between cyanobacteria and algae are important in the 550 nm region, whereas PC absorption near 620 nm and Chl-*a*-related absorption and fluorescence features near 680 nm are also highly important. The most important spectral regions for CAR determination (or cyanobacteria discrimination) were between 525 and 575 nm and 660 to 690 nm. Unsurprisingly, this is very similar to the ranges identified for PC and relates to (1) shifts in the green peak caused by variable accessory pigmentation and (2) Chl-*a* fluorescence-related features. For cyanobacteria discrimination, three bands positioned near 548, 674, and 704 nm might be sufficient when using *R_rs_*.

#### 3.1.2. Band Positions for TOAR and BRR

There were significant shifts in the positions of attribution score peaks for TOAR and BRR data compared to *R_rs_*, generally towards higher wavelengths (as the differences in the results for TOAR and BRR were in many cases negligible, these are discussed together and collectively referred to as TOA data types). The factor driving these differences is the attenuation (absorption and scattering) of the water-leaving signal by the atmosphere, which is strongest in the blue due to aerosols. It is apparent that at TOA, there is more dependence on NIR wavebands (above 700 nm), which are less impacted by aerosol contamination. The three primary spectral regions identified as of highest importance for Chl-*a* were 590 to 640 nm, 660 to 700 nm, and, finally, 700 to 740 nm. The spectral range below 600 nm is of little importance. It is evident that there is a large, almost continuous region from 660 to 740 nm that is most important for Chl-*a* determination, with the most important region just beyond the red edge between 710 and 740 nm. The shift towards longer wavelengths with increasing Chl-*a* concentration is well known and has been exploited in operational algorithms (e.g., the Maximum Peak Height (MPH) [8] and Adaptive Reflectance Peak Height [23] algorithms). The three most important bands for Chl-*a* were 614, 674, and 722 nm.

For PC, the most important spectral regions are located between 540 and 570 nm, 610 and 660 nm, and 700 and 740 nm. Again, there is a shift in feature importance attributions towards the red. Most strikingly, however, is the absence of the 660 to 690 nm attribution peaks visible in *R_rs_*, corresponding to the location of the maximum effects of Chl-*a* absorption and fluorescence. This may be because subtle fluorescence and absorption effects are obscured at TOA due to atmospheric effects. In contrast, the NIR region between 700 and 740 nm was more important for PC detection, which is likely related to the signal produced by the scattering of cyanobacterial cells [24].

For CAR, the two most important spectral regions at *R_rs_* were also shifted towards the red, located between 540 and 600 nm and 700 and 740 nm. Most interesting is that cyanobacteria discrimination can be determined with spectral regions below 600 nm and above 700 nm, seemingly ignoring important spectral markers between 600 and 700 nm. The 660 to 690 nm region, where effects from Chl-*a* absorption and fluorescence are highest, was of little importance for cyanobacteria discrimination at TOA. Additionally, the 620 nm PC absorption band had little importance. This again could be evidence that atmospheric attenuation renders subtle effects in PC absorption and Chl-*a* absorption and fluorescence unimportant at TOA. In contrast, it appears that differences in green peak position (540 to 600 nm) and red edge peaks (700 to 750 nm) induced by scattering effects were more distinctive of cyanobacteria blooms than pigment absorption and fluorescence markers.

### 3.2. Product Retrieval for Band Configurations

The retrieval performance from the ML algorithm using the “Full” spectral configuration (all 100 bands) and the P1 and P2 band configurations determined in the previous section is shown in Figure 4. 

For Chl-*a*, PC, and CAR retrieved using *R_rs_*, there was a negligible difference in R^2^ performance between the Full and P1 spectral configurations. However, using more spectral information reduced the error of estimate (lower MAPE and RMSELE values). There was little advantage in using “hyperspectral” resolution over the multi-spectral 10-band configuration with the ML approach. Retrieval performance using only the three most important bands (P2 configuration) had lower R^2^ values and larger errors. However, while utilizing more spectral information reduces error, it is still feasible to provide product estimates with as few as three ideally positioned bands. For example, the minimal P2 spectral configuration may be used to provide initial products in instances where only very few spectral bands are available for download in experimental instrument launches. 

An optimal spectral configuration that provides both sufficient performance for product estimation and data efficiency is likely to consist of around nine well-positioned spectral bands in the 500 to 800 nm range. When comparing the product retrieval performance of TOA data types and those for *R_rs_*, the effect of atmospheric attenuation is apparent. For Chl-*a*, the MAPE increases when using Full resolution TOA data from 10 to 19 ug/L and from 13 to 23 ug/L using the P1 channel configuration. The results were similar for PC, with errors increasing using Full TOA data from 23 to 38 ug/L and from 37 to 47 ug/L with the P1 configuration. The results demonstrate that retrieval of products directly from TOA or using partial atmospheric corrections is feasible, albeit with larger errors than with *R_rs_*.

There was very similar performance in product retrieval for TOAR and BRR data types. The error estimates were roughly equivalent for PC; however, estimates of Chl-*a* and CAR were slightly better with TOAR than BRR. Thus, there was little advantage in using Rayleigh-corrected data with the ML approach. Retrieval performance with TOA data using the P1 configuration was only slightly worse than using Full resolution. There was, however, a significant reduction in performance using the three-channel P2 configuration, where the MAPE almost doubled for Chl-*a* and PC. Cyanobacterial discrimination TOA required more spectral information than three bands can offer, as evidenced by the poor retrieval performance (R^2^ ~ 0.3). These results demonstrate that a P1-type configuration with around nine ideally placed bands provides almost as good performance as Full spectral resolution.

### 3.3. Assessment of 8 nm FWHM on Product Retrieval

The narrower 8 nm FWHM configuration provided a slight improvement in retrieval performance over the wider 12 nm FWHM configuration, as evidenced by slightly smaller MAPE values (Table 2). However, the performance metrics were not unanimous, with variable RMSELE performance and negligible difference in R^2^ values. This suggests that there was no significant advantage in using a narrower 8 nm spectral configuration for pigment concentration estimates. The narrower 8 nm waveband configuration did not improve CAR estimation or cyanobacteria discrimination. We hypothesize that narrower band widths may under-sample spectral features associated with absorption maxima whose widths are closer to 20 nm.

### 3.4. Minimum Viable Spectral Configuration

The minimum viable band configuration for product retrieval is a synthesis of the three most important bands identified by the attribution scores (Figure 5, Table 3). There is considerable overlap and repetition in the P2 band configurations between products and data types. The synthesized P2 configuration gives between eight or nine bands as a minimum viable configuration for product retrieval (the reader should note that algorithm performance for the synthesized spectral configurations was not assessed, as algorithm optimization was not the aim of this investigation.). 

From Figure 3, it is evident that whilst the band centers may differ slightly, four spectral areas are critical to sample:The “green peak” near 540 to 560 nm.The PC absorption and fluorescence-related features between 610 and 660 nm.The Chl-*a* absorption and fluorescence-related features near 670 to 690 nm.The particulate scattering peak between 700 and 730 nm.

It is also evident that there are differences in optimal band position between the *R_rs_* and TOA data types: the most important features for *R_rs_* are centered on absorption features (e.g., PC and Chl-*a* absorption at 626 and 674 nm). These are shifted slightly further towards the red for TOA data types. Therefore, algorithms designed for use with TOA data types require slightly different spectral configurations than those designed for use with *R_rs_*. By combining the P1 spectral configurations, an optimal band configuration that provides product retrieval performance equivalent to Full resolution but using significantly fewer spectral bands is shown in Figure 6 and Table A1. Examining Figure 6, the importance of the red and NIR bands is evident, with the spectrum being sampled almost continuously between 650 and 775 nm. 

## 4. Discussion

### 4.1. Spectral Configuration for Cyanobacteria Discrimination

This study has investigated spectral configurations for a hyperspectral imager for the discrimination of cyanobacterial blooms and the estimation of phytoplankton pigment concentrations with ML. We found that a reduced spectral configuration of around nine ideally positioned spectral bands enabled product retrieval and cyanobacterial discrimination with almost the same accuracy as using all the available spectral channels. This finding is significant, as it provides evidence that with ML, a multi-spectral configuration with ideally positioned bands has almost the same accuracy as a configuration with hyperspectral resolution. This finding is confirmed by [12,13] that showed that, when estimating PC using ML, hyperspectral configurations were sometimes outperformed by multispectral configurations, or had only marginally better performance metrics, depending on the water type and the PC:Chl-*a* ratio. This validates our hypothesis that hyperspectral resolution instruments sampling the entire spectrum are not required for cyanobacterial discrimination and phytoplankton pigment concentration estimation, provided that narrow spectral channels (12 nm nominal FWHM) are ideally positioned. We found that a minimum viable spectral configuration with as few as three spectral bands enabled pigment retrieval (albeit with larger errors), but this was not suitable for cyanobacterial discrimination via the CAR parameter. This demonstrates how accurate cyanobacteria discrimination with multispectral sensors with a few wide bands (e.g., Sentinel-2 and Landsat) has not been achievable, even with ML approaches. Based on our findings, we conclude that cyanobacteria discrimination with ML most likely requires at least five to six narrow bands between 600 and 750 nm. 

The study demonstrates how value-added products can be estimated using TOA data types, albeit with larger errors than with *R_rs_*. Given the difficulty of atmospheric correction over complex inland waters [15,25], this finding highlights the feasibility of approaches based on TOA data types [8,15,25] where the relative errors from atmospheric correction outweigh the advantages of using *R_rs_*. There was no improvement in product performance by using a partial atmospheric correction, even though spectral features may be more discernible after the removal of Rayleigh effects. In almost all cases, parameters were retrieved with equivalent or slightly smaller errors using TOA reflectance compared to BRR. This may be because aerosol effects dominate Rayleigh effects at TOA. Lastly, the narrower 8 nm FWHM configuration did not improve Chl-*a* or PC retrieval and cyanobacteria discrimination compared to the nominal 12 nm configuration, providing evidence that using more narrow spectral bands is unnecessary.

The outcomes for CyanoSat, and hyperspectral sensors in general, are as follows:A spectral configuration with a reduced number of optimally positioned bands can be used for pigment retrieval and cyanobacteria discrimination with ML. This will help to reduce data requirements for download.Algorithms can be applied with TOA data types to avoid errors associated with atmospheric correction.The 12 nm FWHM band configuration is sufficient since there is no or little advantage in using the narrower 8 nm FWHM configuration.As the analysis was performed with every third of CyanoSat’s 300 spectral bands, adjacent bands can be grouped to increase the SNR three-fold without compromising spectral sharpness or product retrieval performance.The methodology employed here could be replicated to prioritize spectral regions for sampling for a broad range of applications.

### 4.2. Limitations

The band configurations presented above do not account for the calculation of “feature interactions” (spectral indices) such as the Fluorescence Line Height [8], the Cyanobacteria Index [7], the MPH [8], or other peak-height measurements [23] or band ratios that are diagnostic of algal or cyanobacterial blooms [14] and have been demonstrated to significantly improve results from ML [12]. These feature interactions are calculated using traditional band positions of ocean color sensors targeting pigment absorption, Chl-*a* fluorescence, and red edge features associated with phytoplankton blooms. Many of these features coincide with the band positions identified for the minimum viable spectral configuration above. As the purpose of this study was not to optimize algorithm performance, but rather to identify the most important spectral features for application with ML, these spectral indices have been neglected. We recommend that the band configurations identified here for cyanobacterial pigments be considered alongside traditional band positions (e.g., those of the Ocean and Land Color Instrument) in future sensors such as Phytoplankton, Aerosol, Cloud and ocean Ecosystem (PACE) (launch 2024), Geosynchronous Littoral Imaging and Monitoring Radiometer (GLIMR) and Surface Biology and Geology (SBG) (planned launch 2027/28) missions. To enable the calculation of simple yet effective spectral discrimination indices and to improve product retrieval performance, we recommend continuous sampling of the red/NIR between 600 and 750 nm (viz. the optimal spectral configuration). 

This study has not optimized algorithm performance beyond the identification of the most important spectral features and simplified band configurations. It is highly likely that by including spectral configurations that enable the calculation of these spectral indices, the retrieval performance would improve. Therefore, the product retrieval performance is only a preliminary indication of the potential performance of the CyanoSat imager. The band recommendations provided here do not account for the avoidance of any atmospheric absorption features in the manner of planned or existing satellite sensors. However, atmospheric effects are by nature taken into account in the assessment at TOA. Further, the band configurations do not consider requirements for atmospheric correction. 

The results do not account for noise introduced by the CyanoSat imager at TOA. Further, the results from *R_rs_* assume the perfect removal of the atmosphere and therefore do not represent the performance of CyanoSat using atmospherically corrected data. Lastly, the results shown are average values across seven different water types, and it is important to note that product performance varies widely according to water type. The spectral configurations identified here have been determined for the specifications of the CyanoSat imager and therefore may not be directly applied to other imagers. Finally, the results shown here use modeled data; however, successful practical applications of product retrievals using ML algorithms to common multispectral sensors, trained on the same synthetic dataset, can be seen in [12]. 

## Figures and Tables

**Figure 1 sensors-23-07800-f001:**
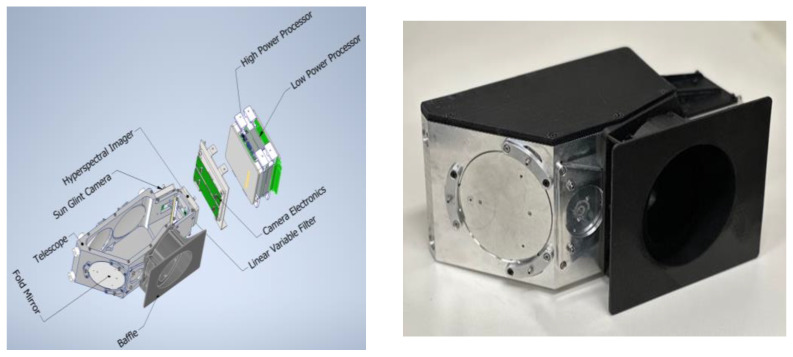
The CyanoSat hyperspectral imaging payload. (**Left**): An isometric view of the payload, highlighting the optomechanical and electrical subsystems. (**Right**): The flight-model optical payload during pre-launch qualification.

**Figure 2 sensors-23-07800-f002:**
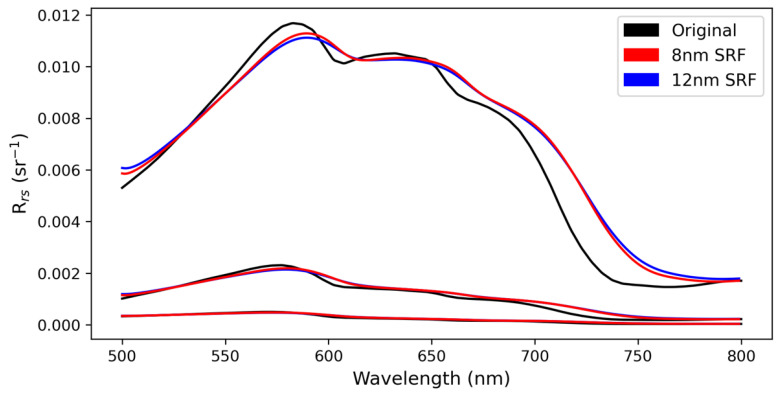
The impact of spectral over-sampling for 300 CyanoSat spectral bands for 8 and 12 nm configurations for three different *R_rs_* spectra representing highly sedimented waters (top), mildly blooming waters (middle), and more oligotrophic waters (bottom).

**Figure 3 sensors-23-07800-f003:**
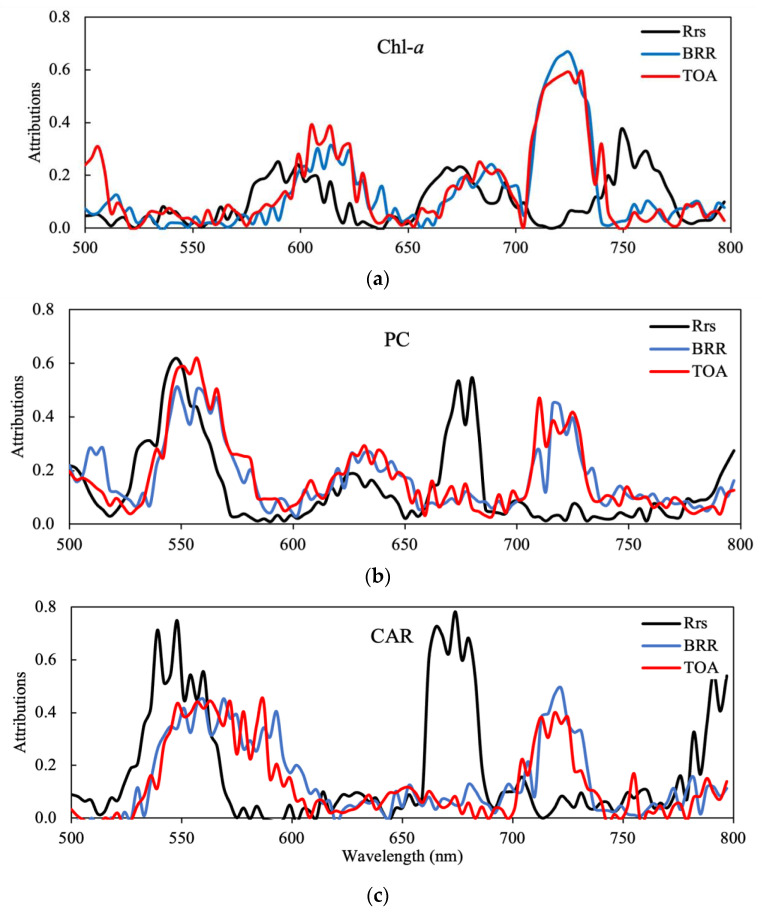
Feature attribution scores in each of the 100 CyanoSat spectral bands for *R_rs_*, TOA, and BRR products. (**a**) Chl-*a*; (**b**) PC; (**c**) cyanobacteria:algae ratio.

**Figure 4 sensors-23-07800-f004:**
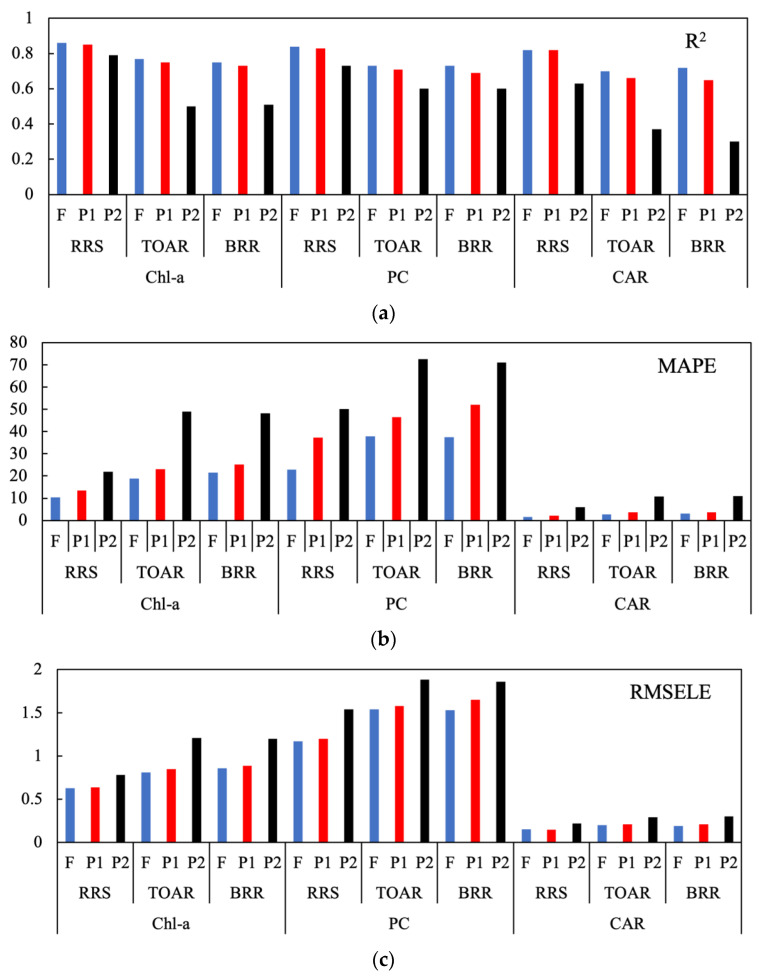
Statistical results for product retrieval using Full (F), P1, and P2 spectral configurations with *R_rs_* (RRS), TOAR, and BRR data types. (**a**) R^2^; (**b**) MAPE; (**c**) RMSELE. Panels (**a**,**b**) use the same x-axis shown in panel (**c**).

**Figure 5 sensors-23-07800-f005:**
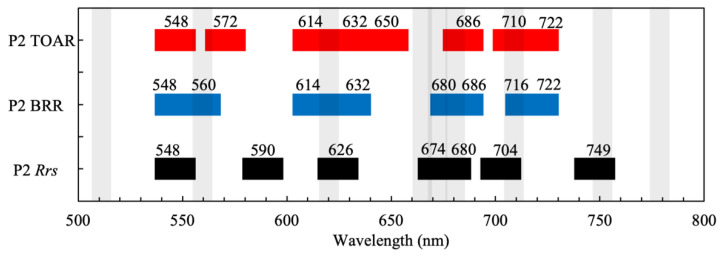
Minimum viable band configuration for TOAR (red), BRR (green), and *R_rs_* (blue) data types at a 12 nm FWHM. Note: adjacent bands may overlap. The overlay shows the approximate positions of OLCI bands (without atmospheric bands).

**Figure 6 sensors-23-07800-f006:**
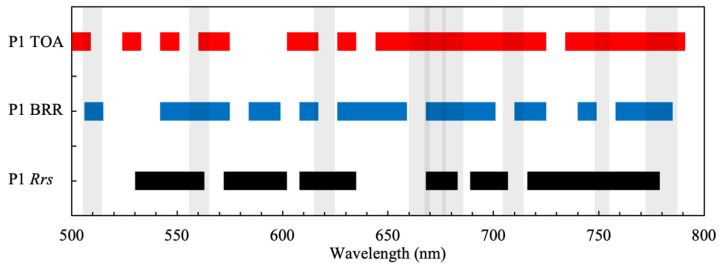
Optimal band configuration using P1 configurations for TOA, BRR, and *R_rs_*. Note: adjacent bands overlap considerably. The overlay shows the approximate positions of OLCI bands (without atmospheric bands).

**Table 1 sensors-23-07800-t001:** P1 and P2 spectral configurations for each product with *R_rs_*, TOAR, and BRR data types.

	Chl-*a*			PC			CAR		
	*R_rs_*	TOAR	BRR	*R_rs_*	TOAR	BRR	*R_rs_*	TOAR	BRR
**P2**	590	614	614	548	548	548	548	572	560
	674	686	686	626	632	632	674	650	680
	749	722	722	680	710	716	704	722	722
**P1**	536	506	512	548	548	512	548	548	560
	590	530	572	578	608	548	560	572	590
	599	566	614	626	632	560	614	614	638
	614	614	638	680	662	596	632	632	656
	623	656	674	698	698	632	674	650	680
	674	686	686	728	710	644	704	674	698
	695	722	722	746	740	680	722	722	722
	749	764	764	764	758	716	740	752	770
	761	782	782	776	770	746	758	788	782

**Table 2 sensors-23-07800-t002:** Performance metrics for 12 and 8 nm FWHM configurations using *R_rs_*.

Product	FWHM	R^2^	MAPE	RMSELE
Chl-*a*	12	0.86	10.3	0.63
	8	0.85	9	0.66
PC	12	0.84	22.9	1.17
	8	0.85	19.5	1.14
CAR	12	0.82	1.57	0.15
	8	0.82	1.72	0.15

**Table 3 sensors-23-07800-t003:** Minimum viable band configuration (P2) according to data type. Bands that appear more than once for a product or data type are shown in bold.

Band No.	TOAR	BRR	*R_rs_*
1	**548**	**548**	**548**
2	572	560	590
3	**614**	**614**	626
4	**632**	**632**	674
5	650	**680**	**680**
6	**686**	686	704
7	710	716	749
8	**722**	**722**	

## Data Availability

The synthetic dataset is available from the authors.

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
