# Peer review of "Determining the Spectral Requirements for Cyanobacteria Detection for the CyanoSat Hyperspectral Imager with Machine Learning"

_sensors, 2023, doi:10.3390/s23187800_

Round 1

Reviewer 1 Report

The manuscript by Matthews et al presents a study of sensor design using simulated spectral data and a simple neural network as an estimator to derive estimates of cyanobacterial abundance to determine an optimal multispectral configuration. I commend the authors for their brevity and clarity of presentation. In general, I found that this manuscript leans a too heavily on the citation #4, and would be good to slightly decouple them slightly by providing some more context. I would suggest though to present more detail of the machine learning setup which is at the heart of the methods. Since algorithm optimization is not an aim of this study, it would be possible to present the setup briefly with an extra para or two in the methods.

The discussed results of the study are excellent, shedding light on the shortcomings of previous ML efforts on wide-band satellite products such as Sentinel for cyanobacterial discrimination. Despite the developed and validated spectral config analysis, the suggestion to "not prefer this spectral configuration over the traditional band positions" (L381) was a surprising volte-face. While a honest assessment is wlecome, I recommend the authors to conclude with some comments about what exactly this study's results CAN be used for. Is it only limited to creating a new CyanoSat multispectral config?

Specific comments
==================

L67: What does "3U visible" mean?

L70: typo? ground sampling distance is not 50 m

Figure 1: Please provide a more detailed caption. Description in 2.1 is missing the physical weight and volume of the imager also.

L135: The setup & config of the MLP algorithm is not described here at all, and in the reference [4] it is somewhere in the supplementary material. As 260k synthetic spectra were used, with varying spectral configs (100 vs P1 vs P2), please provide some detail in the MLP training and validation here. This will ease readability and in assessing the results presented.

L160: What was the error measure L(y, f) that was used? How would feature importance rank differently if one of the other errors was used?

L164-179: These sound like results in the methods section, but as I understand it the intent is to downsample the spectral bands and also to choose test spectral configs P1 and P2. Would be useful to include some text at the start of the para to clarify this.

L170-179: How were the spectral bands for P1 and P2 chosen? This needs explaining. were only top 9 or top 3 ranked bands in terms of feature importance used? The traces in Fig 2 would suggest that the resulting spectral bands would not be as widely spread as presented in the table.

Table 2 is rather difficult to read and compare. I wonder if it would not be better to present these also as figure(s), with one panel each for the error measures (R2, MAPE, RMSELE) and bar graphs for the factorial results.

Table 4: No entries are in boldface, as suggested in the caption.

L355: typo in "This may be because aerosol effects dominate aerosol effects at TOA"

Author Response

The manuscript by Matthews et al presents a study of sensor design using simulated spectral data and a simple neural network as an estimator to derive estimates of cyanobacterial abundance to determine an optimal multispectral configuration. I commend the authors for their brevity and clarity of presentation. In general, I found that this manuscript leans a too heavily on the citation #4, and would be good to slightly decouple them slightly by providing some more context. I would suggest though to present more detail of the machine learning setup which is at the heart of the methods. Since algorithm optimization is not an aim of this study, it would be possible to present the setup briefly with an extra para or two in the methods.

The discussed results of the study are excellent, shedding light on the shortcomings of previous ML efforts on wide-band satellite products such as Sentinel for cyanobacterial discrimination. Despite the developed and validated spectral config analysis, the suggestion to "not prefer this spectral configuration over the traditional band positions" (L381) was a surprising volte-face. While a honest assessment is wlecome, I recommend the authors to conclude with some comments about what exactly this study's results CAN be used for. Is it only limited to creating a new CyanoSat multispectral config?

The Authors would like to thank the reviewer for their time to put together a constructive review.

We have revised the sentence from the negative to a positive statement as follows:

“We recommend that the band configurations identified here be considered alongside traditional band positions (e.g., those of the Ocean and Land Color Instrument) in future sensor configurations.”

Added to abstract: “The spectral configurations identified here could be used to guide the selection of bands for future ocean and water color radiometry sensors.”

To highlight how the results can be used we have amended line 381 as follows and added an additional bullet point:

“The outcomes for CyanoSat, and hyperspectral sensors in general, are as follows:

  • The methodology employed here could be replicated to prioritize spectral regions for sampling for a broad range of applications

Specific comments
==================

L67: What does "3U visible" mean?

Removed 3U which is a technical engineering term.

L70: typo? ground sampling distance is not 50 m

Changed to spatial resolution.

Figure 1: Please provide a more detailed caption. Description in 2.1 is missing the physical weight and volume of the imager also.

These have been added as follows:

“The CyanoSat hyperspectral imaging payload. Left: An isometric view of the payload highlighting the optomechanical and electrical subsystems. Right: The flight-model optical payload during pre-launch qualification.”

Line69: “To ensure compatibility with cube-sat class satellite buses the payload has a volume envelope of 100 x 100 x 300 mm and a weight of 2.7 kg.”

L135: The setup & config of the MLP algorithm is not described here at all, and in the reference [4] it is somewhere in the supplementary material. As 260k synthetic spectra were used, with varying spectral configs (100 vs P1 vs P2), please provide some detail in the MLP training and validation here. This will ease readability and in assessing the results presented.

We have added further information on the training, validation and architecture of the algorithm as follows:

“A detailed overview of how the algorithm was configured is found in [4], but briefly described here. First, the hyperspectral synthetic data was resolved to the different hypothetical band configurations using the sensor spectral response functions. The MLP model used for this study was composed of an input layer, which accepted the visible and near infrared channel reflectances of the specific sensor configuration, followed by five hidden layers of 100 neurons each and an output layer which included a singular water parameter of choice. The activation function, which maps the summed weighted inputs to the output of the neuron, was performed using Rectified Linear Units (ReLu). The Adam optimization algorithm was used to iteratively update network weights based on training data and acts as a more functional extension of stochastic gradient descent, where the loss function used was mean absolute error (MAE). Several feature transformations were performed on the data before training. The input channel reflectances were log transformed, as well as output chl-a and PC concentrations. Input reflectances were then standardized by removing the mean and scaling to unit variance. The CAR data was untransformed. The dataset was split into 80% for training and k-fold cross validation for five folds was used to avoid sampling bias, and 20% of the data was used for validation and application of statistical performance metrics.”

L160: What was the error measure L(y, f) that was used? How would feature importance rank differently if one of the other errors was used?

An explanation has been added as follows:

“R2 was used as the error metric for feature permutation. Other error metrics were tested, however, showed quite similar results and thus R2 was used due to its interpretability in regression problems and sensitivity to model perturbations. Future work could potentially take a band-wise average of multiple error metrics, but this would vastly increase computation time and would not be necessary for a theoretical exploration such as this.”

L164-179: These sound like results in the methods section, but as I understand it the intent is to downsample the spectral bands and also to choose test spectral configs P1 and P2. Would be useful to include some text at the start of the para to clarify this.

We have added text explaining feature independence and explaining how and why the configurations were chosen (see below).

L170-179: How were the spectral bands for P1 and P2 chosen? This needs explaining. were only top 9 or top 3 ranked bands in terms of feature importance used? The traces in Fig 2 would suggest that the resulting spectral bands would not be as widely spread as presented in the table.

We have added the following explanations in the text:

“The first spectral configuration, called "P1”, consists of nine attribution peaks. This number was chosen after experimenting with different configurations and finding that nine bands is close to the optimal spectral configuration that enables product retrieval with almost the same accuracy as Full resolution but using significantly fewer bands. Very little difference was found in performance for 12, 15, 20 or more bands.Secondly, a “P2” configuration was chosen consisting of the three primary spectral attribution peaks. Three bands were decided on after noticing that three broad regions of importance could be defined from the feature important plots for each product.”

Table 2 is rather difficult to read and compare. I wonder if it would not be better to present these also as figure(s), with one panel each for the error measures (R2, MAPE, RMSELE) and bar graphs for the factorial results.

On the reviewer’s advice we have converted the table into a figure (please see revised manuscript).

Figure 3. Statistical results for product retrieval using Full (F), P1 and P2 spectral configurations with Rrs(RRS), TOAR and BRR data types. (a) R2; (b) MAPE; (c) RMSELE. Panels (a) and (b) use same x-axis shown in panel (c).

Table 4: No entries are in boldface, as suggested in the caption.

Corrected.

L355: typo in "This may be because aerosol effects dominate aerosol effects at TOA".

Corrected.

Reviewer 2 Report

Dear authors, I am not an expert in cyanobacteria, but I do have a lot of experience in using ML techniques with multi-spectral imaging. In that context, I think it is essential to add transparency to the predictions of ML models, and not to use them as black boxes to be blindly trusted. Therefore, I found your work interesting and useful. However, some issues should be better explained:

First, techniques based on introducing perturbations have an important limitation: they assume that the inputs are independent. However, they are not. It does not make much sense to modify one band without modifying nearby bands. Therefore, the generated data, with which you are going to evaluate the importance of each band, are not realistic. Perhaps it could be improved by introducing perturbations in a region centered on each band. That is, instead of modifying band n, modify the bands from n-k to n+k. Other simple techniques can also be tried. Reading your previous work, I see that you have used decision trees for this same problem. If you have used GBDT, or other similar ones, the libraries themselves provide metrics about the importance of each feature. For example, with GBDTs you have the function “feature_importance”.

In any case, looking at the results, it does not seem that this lack of realism has generated major problems.

Secondly, the paper proposes from the beginning two configurations: P1 and P2 with 9 and 3 bands respectively. Why have you chosen these numbers? It is not intuitive. I understand that there is a process behind it, but it is not well explained. I think it would be clearer if an analysis is done by progressively reducing the number. From 100 you can go to 32, 16... And from that analysis you select those configurations P1, and P2. Please justify why you use these numbers, and/or extend the analysis to other values.

Author Response

Dear authors, I am not an expert in cyanobacteria, but I do have a lot of experience in using ML techniques with multi-spectral imaging. In that context, I think it is essential to add transparency to the predictions of ML models, and not to use them as black boxes to be blindly trusted. Therefore, I found your work interesting and useful. However, some issues should be better explained:

First, techniques based on introducing perturbations have an important limitation: they assume that the inputs are independent. However, they are not. It does not make much sense to modify one band without modifying nearby bands. Therefore, the generated data, with which you are going to evaluate the importance of each band, are not realistic. Perhaps it could be improved by introducing perturbations in a region centered on each band. That is, instead of modifying band n, modify the bands from n-k to n+k. Other simple techniques can also be tried. Reading your previous work, I see that you have used decision trees for this same problem. If you have used GBDT, or other similar ones, the libraries themselves provide metrics about the importance of each feature. For example, with GBDTs you have the function “feature_importance”. In any case, looking at the results, it does not seem that this lack of realism has generated major problems. 

The authors would like to thank the reviewer for taking the time to put together a constructive review. We have attempted to address your concerns regarding independence and also added extra details regarding the ML model in response to another reviewer's comments (please refer to revised manuscript to see all changes that have been made). 

The following explanation has been added to the text:

Feature permutations on spectral data come with certain caveats. Permutation importance function under the assumption that each feature (or band in this case) is independent of one another, whereas with spectral data, adjacent bands might be correlated, and feature importance could be misestimated or misinterpreted. To examine this effect, tests using all channels showed very little perceptible difference between adjacent channels (at an effective 1 nm incremental spacing) in attribution scores, and product retrieval performance using all 300 channels was nearly identical to that using every second or third channel (200 or 100 channels).”

Secondly, the paper proposes from the beginning two configurations: P1 and P2 with 9 and 3 bands respectively. Why have you chosen these numbers? It is not intuitive. I understand that there is a process behind it, but it is not well explained. I think it would be clearer if an analysis is done by progressively reducing the number. From 100 you can go to 32, 16... And from that analysis you select those configurations P1, and P2. Please justify why you use these numbers, and/or extend the analysis to other values.

The following justification for the spectral configurations have been added to the text:

“The first spectral configuration, called "P1”, consists of nine attribution peaks. This number was chosen after experimenting with different configurations and finding that nine bands is close to the optimal spectral configuration that enables product retrieval with almost the same accuracy as Full resolution but using significantly fewer bands. Very little difference was found in performance for 12, 15, 20 or more bands.Secondly, a “P2” configuration was chosen consisting of the three primary spectral attribution peaks. Three bands were decided on after noticing that three broad regions of importance could be defined from the feature important plots for each product.”

Other changes:

Figure 3 and 4 have been re-drawn to include color and also show the position of OCLI band positions for reference to the discussion.